# The Effect of Sensitization on the Susceptibility of AISI 316L Biomaterial to Pitting Corrosion

**DOI:** 10.3390/ma16165714

**Published:** 2023-08-21

**Authors:** Viera Zatkalíková, Milan Uhríčik, Lenka Markovičová, Lucia Pastierovičová, Lenka Kuchariková

**Affiliations:** Department of Materials Engineering, Faculty of Mechanical Engineering, University of Žilina, Univerzitná 8215/1, 010 26 Žilina, Slovakia; milan.uhricik@fstroj.uniza.sk (M.U.); lenka.markovicova@fstroj.uniza.sk (L.M.); lucia.pastierovicova@fstroj.uniza.sk (L.P.); lenka.kucharikova@fstroj.uniza.sk (L.K.)

**Keywords:** AISI 316L biomaterial, sensitization, pitting corrosion, grain boundaries

## Abstract

Due to the combination of high corrosion resistance and suitable mechanical properties, AISI 316L stainless steel is extensively used as the biomaterial for surgical implants. However, heat exposure in inappropriate temperatures can cause its sensitization accompanied by chromium depletion along the grain boundaries. This study deals with an assessment of the susceptibility of sensitized AISI 316L biomaterial to pitting under conditions simulating the internal environment of the human body (Hank’s balanced salt solution, 37 ± 0.5 °C). The resistance to pitting corrosion is tested by the potentiodynamic polarization and by the 50-day exposure immersion test. Corrosion damage after the exposure immersion test is evaluated in the specimens’ cross-sections by optical microscope and SEM. Despite passive behavior in potentiodynamic polarization and shallow, slight corrosion damage observed after exposure, the sensitized AISI 316L biomaterial could represent a risk, especially in long-term implantation even after the chemical removal of high-temperature oxides.

## 1. Introduction

AISI 316L is the Cr-Ni-Mo grade of austenitic stainless steel (SS) widely used in the medical field preferred for its excellent corrosion resistance and appropriate mechanical properties [1,2,3,4,5]. Due to the high-quality passive surface film, AISI 316L exhibits exceptional resistance to general corrosion in oxidation environments [6,7,8,9]. Molybdenum addition suppresses a susceptibility to the localized corrosion, such as pitting and crevice corrosion in chloride environments [10,11]. This is particularly important in medical implant applications as it ensures the longevity and reliability of the implant when exposed to the corrosive environment of the human body. AISI 316L is considered biocompatible because it is well-tolerated by the human body and minimizes adverse biological responses [1,2,4,5]. It has been extensively used in surgical implants, such as orthopedic implants (e.g., screws, plates, and joint replacements), cardiovascular implants (e.g., stents), and dental implants. AISI 316L offers high tensile strength and toughness and this makes it suitable for load-bearing applications where the implant needs to withstand mechanical stresses and forces without deformation or failure. AISI 316L can be easily fabricated into complex shapes and designs. It can be formed, welded, and machined using common fabrication techniques, allowing for the production of customized implants that meet the specific requirements of individual patients [2,5]. It exhibits good resistance to fatigue failure, (tendency of a material to fail under repeated cyclic loading) which is important for components and implants subjected to cyclical stresses [2,5]. AISI 316L is known for its low magnetic permeability, which is a valuable property in various medical applications, particularly in the field of diagnostic imaging (e.g., Magnetic Resonance Imaging (MRI)).

The corrosion resistance of austenitic stainless steels is compromised in the case of precipitation of chromium carbides at the grain boundaries. Due to this phenomenon, the austenitic grains become locally chromium-depleted. If the proportion of chromium is reduced below the passivation limit, sensitization with consequent susceptibility to the intergranular corrosion in aggressive environments occurs [12,13,14,15,16,17]. The mentioned process, resulted from the exposure to the so-called critical temperatures (500 °C to 800 °C) with slow cooling may most often come into consideration at an inappropriately chosen processing temperature. The minimum time for sensitization depends on several factors including the specific alloy composition, temperature, carbon content, heating rate and presence of other alloying elements [12,13]. It is important to note that sensitization is a gradual process that occurs over time. The conditions for the significant extensive sensitization of austenitic SS can be determined according to the diagram of carbon solubility in austenite [18,19]. However, under specific conditions local sensitization can be related to a short-term heating, e.g., during welding [20,21]. In this case, the heat-affected zones of the SS weldments show clear signs of local sensitization. According to the authors of [22], a sensitization of austenitic SS can also occur after an exposure to the critical temperatures caused by a fire depending on its temperature, intensity and duration.

Local sensitization also applies to biomedical applications. Welding allows for the creation of strong and durable bonds between components. In the case of stainless steel implants, specialized welding techniques and processes are employed to ensure the integrity of the final complex-shaped implants (e.g., orthopedic screws, joint replacements, implantable cardiac rhythm devices, cochlear implants, neurostimulators) to join their multiple components together [23,24,25,26].

The authors of [27] studied boundary chromium depletion and degree of sensitization of 301 austenitic SS and found a dependence of these phenomena on the austenitic grain size. The authors of [28] recorded that an increase in nickel content significantly supports formation of chromium carbides by reduction of carbon solubility in austenite. From the kinetic point of view, nickel affects activities of carbon and chromium.

Despite the low carbon content, sensitization also occurs with AISI 316L stainless steel and induces a higher susceptibility to the pitting corrosion [29,30,31,32]. According to the authors of [25,33,34,35], chromium-depleted zones along the grain boundaries more susceptible to corrosive attack compared to the surrounding matrix become preferential sites for pit initiation. Once pitting corrosion is initiated at the sensitized grain boundaries, it can propagate deeper into the material. Pits can grow in size and depth, compromising the integrity and strength of the stainless steel. This can result in premature failure, especially in environments where pitting corrosion is prevalent, such as chloride-rich solutions [8,9,36,37].

Most current studies [17,30,31,32,33,34,35] evaluate the effect of sensitization on the resistance to the pitting corrosion of AISI 316L steel by electrochemical methods (often by potentiodynamic polarization and EIS) in NaCl solutions (0.01–3.5%). The above-mentioned authors record a significantly higher susceptibility to pitting and degraded quality of the passive surface film. The authors Farooq et al. [25] evaluate the pitting corrosion resistance of sensitized 316L steel in simulated physiological solutions, but do not consider human body temperature, which can be a significant factor of the corrosion resistance.

The subject of this research is the assessment of the susceptibility of sensitized AISI 316L biomaterial to pitting under conditions simulating the internal environment of the human body (Hank’s balanced salt solution, 37 ± 0.5 °C). The resistance to the pitting corrosion is tested by the potentiodynamic polarization (PP) and a 50-day exposure immersion test. Corrosion damage after the exposure immersion test is evaluated in the cross-sections of the specimens in the as-received state and after sensitization by optical microscope (OM) and SEM.

## 2. Materials and Methods

The material used for the experiments was AISI 316L SS in a sheet of 1.5 mm thickness. It was produced by continuous casting in an electric arc furnace and consequently annealed at 1050 °C. The resulting smooth surface (2B surface finish) was ensured by cold rolling and pickling. The composition expressed as weight percentages of the elements (obtained by X-ray fluorescence) is presented in Table 1.

The specimens of rectangular shape (dimensions 15 mm × 40 mm × 1.5 mm) were prepared for the corrosion resistance tests. Part of the specimens were used in the original as-received state. The additional specimens were used for heat exposure carried out before the corrosion tests.

The heat exposure for sensitization was carried out in a furnace at 650 °C for 40 h. The temperature and the heating time were adjusted for an extensive sensitization of the tested material according to the diagram of carbon solubility in austenite presented in studies [18,19] (a local sensitization reached by a simulation of the welding conditions was not the aim of this study). After finishing the heating, the specimens were cooled in the turned-off furnace to ensure slow consequent cooling in the air. This created suitable diffusion conditions for chromium carbide precipitation. For the confirmation of the specimen sensitization, the oxalic acid etching test carried out by the A practice of ASTM A262 standard method [38] was used. The conditions applied for the test (solution composition, temperature, time, current density) are listed in Table 2. The oxalic acid test was carried out on the metallographically prepared, ethanol-rinsed and air-dried surface. During electrochemical etching, the tested specimen was connected to the positive pole as the anode (+); the stainless steel block was used as the cathode (−). The cathode and anode were parallel with distance approx. 5 mm. The etched surface was evaluated by OM [38].

Before the corrosion tests, a part of the sensitized specimens were chemically treated for removal of the high-temperature oxides which generally reduce corrosion resistance of the heat-affected stainless steels [39]. The conditions of the applied chemical surface treatment are presented in Table 3. The treatment solution was chosen to contain nitric acid (HNO_3_), which is used as an effective oxidizing agent to strengthen the passive surface film and to raise the redox potential [8,9,32,39]. Hydrofluoric acid (HF) generates H^+^ ions and it also acts as complexing agent for both Fe^3+^ and Cr^3+^ ions [32]. The other sensitized specimens were left as the “heat tinted” and used for comparison in potentiodynamic polarization tests.

An overview of tested specimen types is given in Table 4.

Hank’s balanced salt solution (pH 7.25, measured before the tests) at a temperature of 37 ± 0.5 °C for simulation of the internal environment of the human body was used as the corrosion environment for both PP and exposure immersion tests. All chemical compounds used for solution were analytical grade.

The potentiodynamic polarization was performed in the conventional three-electrode cell system with a calomel reference electrode (SCE, +0.248 V vs. SHE at 20 °C) and a platinum auxiliary electrode (Pt) using Bio-Logic corrosion measuring system with PGZ 100 measuring unit. The time for potential stabilization between the specimen and the electrolyte was set to 10 min. The exposed area of a specimen was 1 cm^2^ (the rectangular specimen (15 mm × 40 mm × 1.5 mm) was externally attached to a 1 cm^2^ round opening on the corrosion cell).

The potentiodynamic polarization curves were recorded at the sweep rate of 1 mV/s [40]; a potential scan range was applied between −0.30 and 1.2 V vs. open circuit potential (OCP). At least three experiment repeats were carried out for each specimen type and the representative curve was selected.

The as-received and sensitized specimens of rectangular shape (15 mm × 40 mm × 1.5 mm) were used for a 50-day exposure immersion test. The specimens were degreased by ethanol and weighted out with accuracy ±0.00001 g before the test. The group of three parallel specimens was tested. For the exposure test, the as-received and sensitized specimens without high-temperature oxides (S type) were used. After the completed test, the specimens were brushed, rinsed by de-mineralized water, freely dried and weighted out again [41].

## 3. Results and Discussion

The microstructure of the as-received material in longitudinal section (observed by OM) is shown in Figure 1. It consists of polyhedral variously oriented austenitic grains with frequent twins, associated probably with annealing or rolling. The microstructure contains almost no observable inclusions.

The performed heat exposure (650 °C/40 h)-induced sensitization confirmed by the microstructure in Figure 2 was obtained by the oxalic acid etching test according to ASTM A262 practice A standard. This standard method enables a quick assessment of sensitization of austenitic stainless steels by the assessment of the grain boundaries attack. If at least one grain is completely surrounded by ditches, the microstructure is classified as the ditch one and it indicates the sensitization of the SS. According to this classification, the state without sensitization manifests itself in the step microstructure—steps between the grains are caused by their various crystallographic orientation. As can be seen in Figure 2, the most austenitic grains are surrounded by ditches arising from carbide dissolution during performed electrochemical etching and this points to the extensive sensitization of the material [19,38,42]. The similar ditch microstructures obtained under comparable sensitization conditions were also observed by the authors [29,43].

### 3.1. Potentiodynamic Polarization

The effect of sensitization on the behavior of AISI 316L SS in potentionynamic polarization is documented in Figure 3. The polarization curves for sensitized state (surface after chemical removal of high-temperature oxides, S curve) and for the as-received state are similar in shape; both curves have passive anodic branches which points to the control of anodic dissolution rate by passive current density, not by corrosion current density. Therefore, these curves were reviewed by the pitting potentials E_p_ (Table 5), which denote a breakdown of the passive surface film and the onset of stable pit growth. The higher E_p_ value reflects the better passive film quality and increased resistance to pitting. E_p_ values were identified as the potentials of strong permanent increase in the current density in the passivity region [8,36,37]. E_corr_ values of the above-mentioned curves (Table 5) were determined as the potentials of the transition from the cathodic to the anodic branches. (A shift of E_corr_ in the positive direction points to a higher thermodynamic stability of the material.)

Unlike the as-received and the S curves, the curve for the sensitized state with high-temperature oxides (SO curve) given for comparison in Figure 3 has a different shape, which does not indicate a passive state but active anodic dissolution. In addition to the E_corr_, the corrosion current density i_corr_, which expresses the kinetics of corrosion reactions and determines the corrosion rate, is an important potentiodynamic parameter in this case. Both E_corr_ and i_corr_ values (Table 5) were obtained from Tafel extrapolation using EC-LAB software for this curve.

According to the PP curves (Figure 3), the worst corrosion resistance was proved in the case of sensitization in combination with high-temperature surface oxides (SO curve) arisen from oxidation of chromium during the heating of the material [44]. This process could cause chromium depletion below the “heat tint” and consequently a destruction of the passive film, which lost its protective role [45]. It is important to remember that for a biomaterial in the internal human body environment, the loss of passivity and active anodic dissolution, observed on the SO curve, are unacceptable. Chemical removal of high-temperature oxides by HNO_3_ + HF solution reduced the negative effect of sensitization (curve S in Figure 3). Nitric acid as an oxidizing acid contributed to the release of Fe and Ni from the metal surface in the form of cations that conditioned the reduction of oxygen dissolved in HNO_3_ solution to OH^−^ anions (O_2_ + 2H_2_O + 4e^−^ -> 4OH^−^) [46]. Consequently, the reaction between OH^−^ and the metallic Cr was the first step to the Cr_2_O_3_ formation (Cr + 3OH^−^ -> Cr(OH)_3_; 2Cr(OH)_3_ -> Cr_2_O_3_ + 3H_2_O). The result was the strengthened surface passive film [46] and the S curve typical for a passivating metal with E_p_ and E_corr_ values slightly lower compared to the as-received curve (Table 5). The positive effect of the chemical pickling (HNO_3_ + HF) on passive behavior of AISI 316L welded joints in 1M NaCl solution (shift of E_p_ in positive direction) was also recorded by the authors [32].

The detail of passivity region in linear axes for S and as-received polarization curves is shown in Figure 4. As can be seen, the passive current density for both curves is securely below 0.05 mA/cm^2^ (this value is considered the limit current density for the passive state) and no significant signs of metastable pit growth are observed. However, the passive current density for S curve is higher than for the as-received curve and it tends to increase. This may indicate a lower stability of the passive state from a kinetic point of view related to the sensitization, which manifested despite the strengthening of the passive film by nitric acid.
Kartaman et al. [34] tested the corrosion resistance of sensitized (675 °C) and consequently mechanically polished AISI 316L SS specimens by PP in 0.01wt.% NaCl solution and recorded E_p_ = 0.391 V vs. SCE and also, similarly to us, higher passive current density compared to the as-received specimen. When partially different conditions [34] are taken into account (lower temperature and chloride concentration, different method of removing high-temperature oxides), the value of the E_p_ can be considered comparable to ours (0.36 ± 0.02 V vs. SCE)The impaired passive behavior and higher susceptibility to pitting of the welded AISI 316L SS in Hank’s and Ringer’s physiological solutions were confirmed by Farooq at al. [25].It should be emphasized that the pitting corrosion resistance of stainless steels strongly depends on the temperature. An increase in temperature by just a few degrees and also its fluctuation (e.g., during disease states) can lead to a significant deterioration of corrosion resistance [47]. The elevated temperature affects negatively the self-healing ability of the passive film. Diffusion rate of chloride ions through the passive film increases, making it easier to reach the raw metal surface. This enhanced diffusion contributes to a more rapid depassivation process; the pitting initiation and propagation are accelerated. This consequently leads to the sharp increase in current density [48]. The above-mentioned trend was documented by the authors of [41] who evaluated the corrosion resistance of sensitized AISI 304 steel in 1M acid NaCl solution at the temperatures of 20 and 50 °C. The conditions of sensitization were the same as in our study (650 °C/40 h). On the chemically treated surface (the same conditions as in our study) and at the 50 °C temperature, the authors noted a loss of passive behavior and an active anodic dissolution of the sensitized SS [41].

### 3.2. Exposure Immersion Test

The 50-day exposure immersion test was performed on sensitized (S) specimens and for a comparison also on the as-received ones. Considering that sensitized specimens with high-temperature surface oxides (SO) did not show passive behavior in potentiodynamic polarization, they were not used for this test. After the test was completed, there were no signs of corrosion visible to the naked eye on the tested specimens. However, OM and SEM observation of cross-section edges (Figure 5 and Figure 6) revealed a local corrosion damage and clear differences between the as-received and the sensitized specimens.

According to the microstructure of the sensitized specimen (Figure 5), the corrosion pits were mostly initiated in the grain boundaries weakened by chromium depletion [31,35,36,49]. A loss of chromium decreased the protection ability of the passive film, which facilitated a penetration of chloride anions inside the material [8,9,10]. This confirms the close relationship between the sensitization and the susceptibility to pitting documented by numerous authors [31,35,36,49]. The observed local corrosion damage is shallow because it reaches a depth of approximately 20 µm, which represents only 1.3% of the 1.5 mm sheet thickness. In spite of this fact, it should be taken into account that during a longer contact of the sensitized material with a chloride-containing solution, it will progress to a greater depth along the grain boundaries. This process is accelerated by autocatalytic pH decrease because of the Fe^2+^ hydrolysis resulting in an increase in H^+^ concentration inside the stable-growing pits [8,50].

Only slight local corrosion damage can be observed on the as-received specimen (Figure 6). In this case, an electro/plasma polishing of the surface could help to improve the quality of the surface passive film [51,52].

The average corrosion rates for both as-received and S specimen types calculated from mass losses are listed in Table 6. It should be remembered that when evaluating local pitting corrosion, the corrosion rate is not one of the most essential parameters, but it enables a rough comparison and it helps to create an idea of the kinetics of the corrosion process. According to the presented values, corrosion rates can be considered very low [53]. This result is consistent with the above-documented shallow corrosion damage and with the passive behavior observed in potentiodynamic polarization. It is important to consider that even a slightly higher corrosion rate of the sensitized biomaterial can have significant implications over time, particularly in applications where long-term durability and corrosion resistance are critical, such as in surgical implants [4,5,6].

## 4. Conclusions

Sensitization induced by the heat exposure of the experimental material (650 °C/40 h) was confirmed by “ditch” microstructure (Figure 2) observed after the electrochemical etching carried out according ASTM A262 practice A standard.The potentiodynamic polarization results showed a passive behavior of the sensitized material without high-temperature oxides (S type), but the pitting potential (E_p_) and the corrosion potential (E_corr_) values were slightly lower compared to the as-received state (Table 4). This points to a higher susceptibility of the sensitized specimens to the pitting. According to the potentiodynamic polarization curve, the sensitized material with high-temperature oxides (SO type) showed an active anodic dissolution (Figure 3), which is unacceptable if it is intended for the internal human body environment.During the 50-day exposure immersion test in the simulated physiological solution, the sensitized specimens without high-temperature oxides (S type) were attacked by a local pitting corrosion with the pits situated in the grain boundaries (Figure 5). This confirmed the close relationship between sensitization and the pitting. The observed corrosion damage indicates a possible progress to deeper parts of the material along the grain boundaries.Based on the performed independent experiments, the sensitized AISI 316L biomaterial could represent a risk even after the chemical surface treatment, especially in the case of long-term implantation.

In further research, it would be interesting to focus on the corrosion behavior of AISI 316L biomaterial with different degrees of sensitization after short exposure times to critical temperatures (on the order of minutes), which would simulate welding conditions. To assess the effect of temperature, corrosion tests could be performed not only at the temperature of 37 °C, but also at a common room temperature and at the temperature of 42 °C (to simulate a disease state). It would also be appropriate to assess to what extent the final surface treatment (polishing) will affect corrosion resistance after sensitization.

## Figures and Tables

**Figure 1 materials-16-05714-f001:**
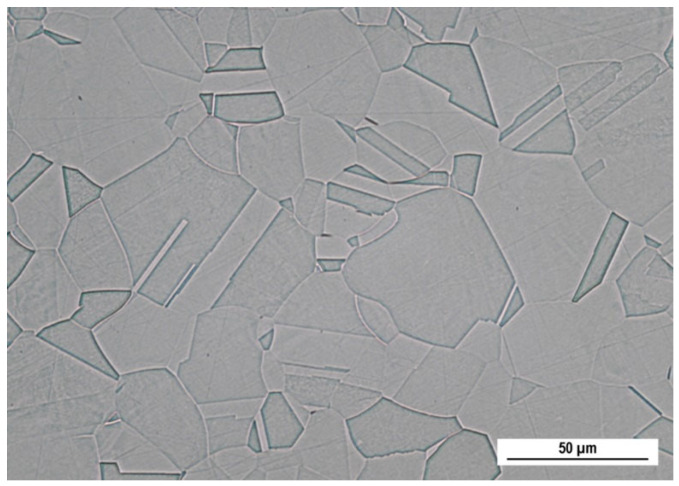
Microstructure of AISI 316L SS, longitudinal section (Kalling’s 2 etch., OM, magnification 500×).

**Figure 2 materials-16-05714-f002:**
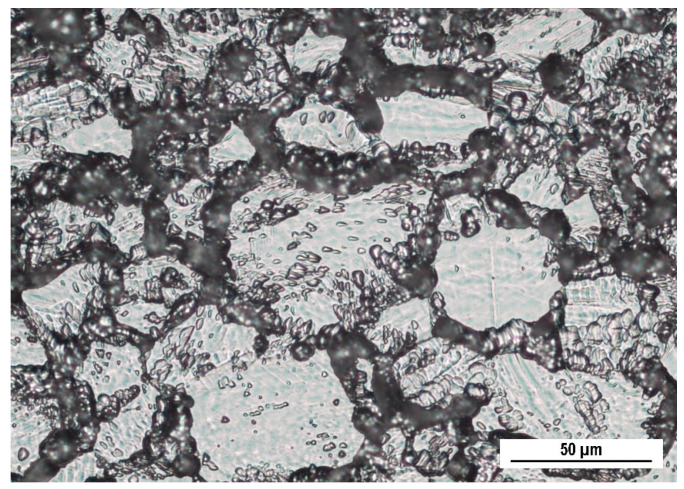
Microstructure of AISI 316L SS after oxalic acid electroetching according ASTM A262 practice A (metallographically prepared surface, OM, magnification 500×).

**Figure 3 materials-16-05714-f003:**
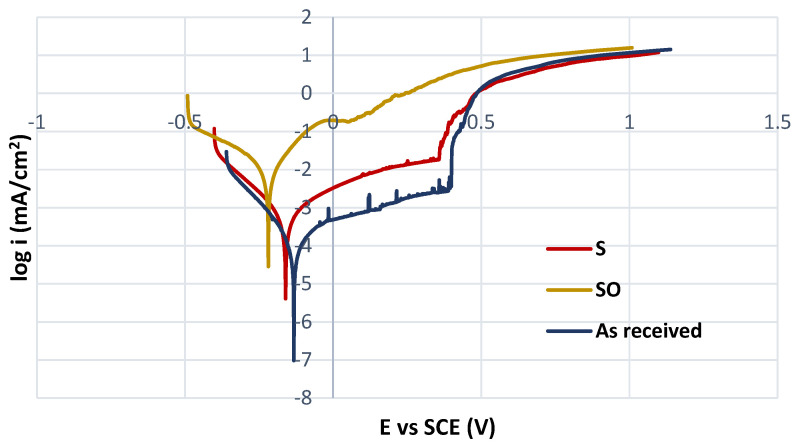
Potentiodynamic polarization curves for three tested specimen types.

**Figure 4 materials-16-05714-f004:**
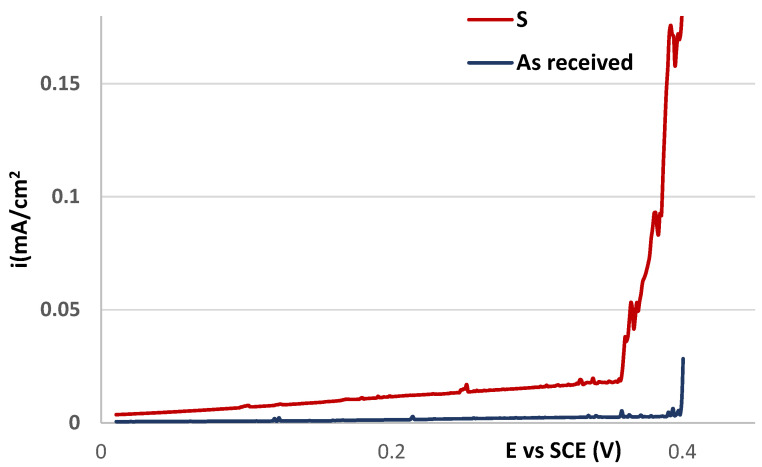
Detail of potentiodynamic polarization curves in linear axes—comparison of sensitized state without high-temperature oxides (S) and the as-received state.

**Figure 5 materials-16-05714-f005:**
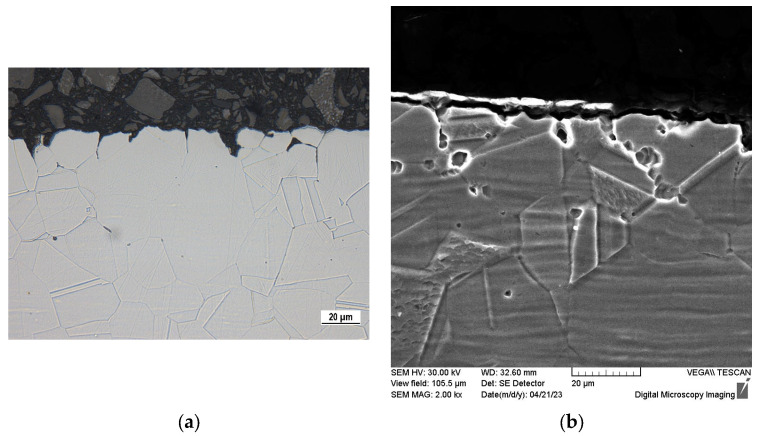
Cross-section edges of sensitized specimen (S) after 50-day exposure in Hank’s balanced salt solution: (**a**) OM, glycerin + HNO_3_ + HF etch., magnification 800×; (**b**) SEM, oxalic acid etch., magnification 2000×.

**Figure 6 materials-16-05714-f006:**
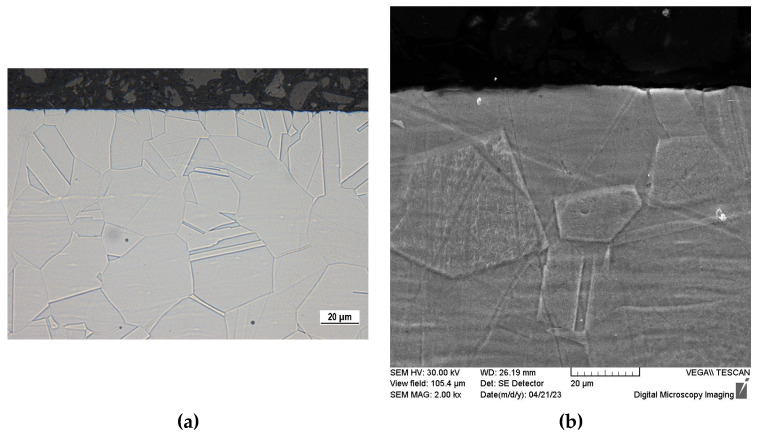
Cross-section edges of as-received specimen after 50-day exposure in Hank’s balanced salt solution: (**a**) OM, glycerin + HNO_3_ + HF etch., magnification 800×; (**b**) SEM, oxalic acid etch., magnification 2000×.

**Table 1 materials-16-05714-t001:** Chemical composition of AISI 316L SS (wt.%).

Cr	Ni	Mo	Mn	N	C	Si	P	S	Fe
16.79	10.14	2.03	0.82	0.05	0.02	0.031	0.03	0.001	balance

**Table 2 materials-16-05714-t002:** Conditions of the electrochemical etching.

Component	Content (wt.%)	Temperature (°C)	Time (s)	Current Density (A.cm^−2^)
oxalic acid demineralized water	10 90	20 ± 3	90	1.0

**Table 3 materials-16-05714-t003:** Conditions of the chemical surface treatment.

Component	Volume (mL)	Temperature (°C)	Time (min)
HF HNO_3_ H_2_O	2 15 to 100 mL	50	10

**Table 4 materials-16-05714-t004:** Overview of tested specimen types.

Type of Specimen	Specimen Designation
Sensitized after removal of high-temperature oxides	S
Sensitized with high-temperature oxides	SO
Original non-treated	As-received

**Table 5 materials-16-05714-t005:** Values of the potentiodynamic polarization parameters.

Specimen Type	Corrosion Potential E_corr_ (V vs. SCE)	Corrosion Current Density i_corr_ (µA/cm^2^)	Pitting Potential E_p_ (V vs. SCE)
As-received	−0.13 ± 0.01	-	0.40 ± 0.02
S	−0.17 ± 0.03	-	0.36 ± 0.02
SO	−0.22 ± 0.04	13.95 ± 1.03	-

**Table 6 materials-16-05714-t006:** Average corrosion rates calculated from mass losses during the exposure test.

Type of Surface	Average Mass Loss (mg)	Average Corrosion Rate (mm/y)
As-received	0.08 ± 0.02	0.0007 ± 0.0001
S	0.11 ± 0.01	0.001 ± 0.0001

## Data Availability

Data sharing is not applicable to this article.

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
