# Peer review of "The Effect of Sensitization on the Susceptibility of AISI 316L Biomaterial to Pitting Corrosion"

_materials, 2023, doi:10.3390/ma16165714_

Round 1
Reviewer 1 Report
In this manuscript the authors study the effect of sensitization of 316L stainless steel on the pitting corrosion resistance in Hank`s solution. The authors conducted electrochemical and immersion tests. Please find my comments below:
1) Authors claim that sensitization of stainless steel parts for biomaterial applications may occur due to welding that is commonly used in the fabrication process. They decided to heat treat their samples at 650C for 40 h. Welding processes last several minutes not 40 h. A better explanation for the long heat treatment might be required.
2) The explanation and discussion on Fig 2 is very brief. Authors are kindly requested to expand the discussion
3) Authors studied 316L SS under three conditions, as received, sensitized with high temperature oxides and sensitized with high temperature oxides removed. What is the most common practise in the biomaterials industry? Are the high temperature oxides usually removed?
4) Authors focus on the pitting corrosion of 316L. Cyclic potentiodynamic polarization curves are needed. The reverse scans vs. the forward scans will help understand which configuration has the best resistance to localized forms of corrosion. Cyclic potentiodynamic polarizations will also help expand the discussion.
5) Figure 4 does not provide anything meaningfull to the discussion. Authors should consider removing it. If their focus is the stability of the passive areas then chronoamperometry (potentiostatic measurements) are more useful
6) Authors should consider enriching Fig 6 with elemental maps to illustrate how thick the surface oxide film is in each configuration and if there is selective dissolution of specific alloying elements.
7) Table 6 is indicative of the resistance to general corrosion of the different configurations of 316L. Nonetheless, the focus of this paper is pitting corrosion. While both configurations show similar corrosion rates after immersion, the resistance to pitting corrosion appears to show big difference.
8) Figure 1 should be in the results rather than in materials and methods
Reviewer 2 Report
Dear Editor,
I am writing to provide my evaluation of the manuscript titled "The effect of sensitization on the susceptibility of AISI 316L biomaterial to pitting corrosion" by Zatkalíková and colleagues. While I find the investigation into the pitting corrosion behavior of sensitized AISI 316L biomaterial to be highly interesting, I believe there are several areas where the manuscript could benefit from improvement. My recommendations are as follows:
The introduction of the paper would benefit from the incorporation of more recent references directly connected to the discussed topics. For example, in the paragraph stating, "...The corrosion resistance of austenitic stainless steels is compromised in the case of precipitation of chromium carbides at the grain boundaries. Due to this phenomenon, the austenitic grains become locally chromium-depleted. If the proportion of chromium is reduced below the passivation limit, sensitization occurs, leading to susceptibility to intergranular corrosion in aggressive environments [12-17]. The mentioned process, resulted from the exposure to the so-called critical temperatures (500 to 800 oC) with slow cooling may come into consideration during welding or at an inappropriately chosen processing temperature..."; it is important to include references such as:
https://doi.org/10.1016/j.engfailanal.2013.01.044
https://doi.org/10.1016/j.engfailanal.2019.104337
https://doi.org/10.1016/j.engfailanal.2015.11.011
These references specifically highlight practical failures resulting from the phenomenon of sensitization in austenitic stainless steels.
I suggest reviewing the following paragraph: "...Degree of sensitization does not correlate directly to the amount of chromium carbide precipitation. It is related to the chromium concentration profiles in the vicinity of the grain boundaries and also to the content of other alloying elements (mostly nickel) that can affect the sensitization kinetics through their influence on the activities of carbon and chromium [22,23]..." This paragraph appears somewhat confusing, and it would be beneficial to ensure that references 22 and 23 support the statements made.
I would like clarification on why the samples were left under open-circuit conditions for only 10 minutes before the polarization measurements. Typically, the open-circuit potential stabilization time for this type of material is approximately 60 minutes. It would be helpful to understand the reasoning behind the shorter stabilization time.
Has any evaluation of the surface of the working electrodes been conducted after the polarization tests? I believe that providing this information would significantly enhance the scientific soundness of the manuscript and improve the understanding of the interrelationships between chemical composition, microstructure, processing, properties, and corrosion performance of the studied materials.
I recommend that the authors review the corrosion current density (icorr) value presented in Table 5 for the SO sample. Upon examining the curve for this material in Figure 3, it appears that the icorr value should be much higher than 7.6 u/cm2.
If possible, I would appreciate it if the authors could provide recommendations for future research and advancements in the field.
Overall, I appreciate the effort put into this research. However, I believe that the manuscript requires further work before it can be considered suitable for publication. Therefore, I recommend accepting the manuscript after major revisions.
Reviewer 3 Report
As stated in my previous report on the article "The effect of sensitization on the susceptibility of AISI 316L biomaterial to pitting corrosion" (Manuscript ID: materials-2513672), this article makes a significant contribution to the understanding of pitting corrosion in sensitized AISI 316L stainless steel under biomedical conditions. The presented results and discussions are robust and provide valuable insights for professionals in the field. Considering the importance of sensitization in the design and use of long-term surgical implants to ensure durability and corrosion resistance, I strongly recommend the acceptance of this article.
The research aims to address the following main question: "What is the susceptibility of sensitized AISI 316L stainless steel to pitting corrosion under conditions simulating the internal environment of the human body?"
The topic is both original and relevant in the field. It explores the specific issue of sensitization and its impact on the corrosion resistance of AISI 316L stainless steel, which is extensively used in surgical implants. This study fills a gap in understanding how sensitization affects the long-term durability and corrosion resistance of this biomaterial, providing valuable insights for biomedical applications.
This research adds significant value to the subject area by providing experimental evidence and comprehensive characterization of the corrosion behavior of sensitized AISI 316L stainless steel. It goes beyond previous studies by simulating physiological conditions and considering the influence of high-temperature surface oxides on corrosion susceptibility. This study contributes to the existing body of knowledge by elucidating the risks associated with sensitization and pitting corrosion in biomedical applications.
The conclusions drawn by the authors are consistent with the evidence and arguments presented in the study. They appropriately address the main question of the research, highlighting the increased susceptibility of sensitized AISI 316L stainless steel to pitting corrosion in simulated physiological conditions. The conclusions emphasize the importance of considering the risks associated with sensitization, even after the removal of high-temperature oxides, in long-term implantation scenarios.
Please find detailed information about the article below:
Introduction:
The introduction provides a good overview of the research topic, emphasizing the significance of sensitization of AISI 316L and its impact on pitting corrosion resistance.
- The first part of the introduction presents a comprehensive introduction to AISI 316L stainless steel as a widely used material in the medical field due to its corrosion resistance and suitable mechanical properties.
- The mention of AISI 316L's excellent general corrosion resistance in oxidizing environments and its ability to withstand localized corrosion, such as pitting and crevice corrosion, in chloride environments, is relevant.
- The specific biomedical applications of AISI 316L, such as orthopedic, cardiovascular, and dental implants, are well mentioned, highlighting its mechanical strength, formability, machinability, and biocompatibility.
- The sensitization of AISI 316L, which can result in chromium depletion along grain boundaries and increased susceptibility to pitting corrosion, is explained clearly.
- Factors influencing the sensitization of AISI 316L, such as chromium concentration profiles near grain boundaries and the presence of other alloying elements, are mentioned.
- The impact of sensitization on pitting corrosion of AISI 316L is emphasized, particularly the formation of chromium-depleted zones along grain boundaries that become preferential sites for pitting initiation.
- Previous studies evaluating the effect of sensitization on pitting corrosion resistance of AISI 316L primarily use electrochemical methods, such as potentiodynamic polarization and EIS, in NaCl solutions.
- The importance of human body temperature in corrosion resistance is not considered in previous studies, justifying the need for this research.
- The objective of this study is to evaluate the susceptibility to pitting corrosion of sensitized AISI 316L biomaterial under conditions simulating the internal environment of the human body, using a balanced saline solution of Hank at a specific temperature.
- The test methods employed, namely potentiodynamic polarization and a 50-day immersion test, as well as the corrosion damage evaluation techniques, including optical microscope observation and SEM, are mentioned.
Section ''2. Materials and Methods'' provides adequate details on the material used, specimen preparation methods, and test techniques employed to evaluate pitting corrosion. The experimental conditions are well described and allow for result reproducibility.
- The section begins by mentioning that the material used is AISI 316L stainless steel with a thickness of 1.5 mm and a smooth and matte surface (2B finish).
- The chemical composition of AISI 316L steel is presented in Table 1.
- The microstructure of AISI 316L steel is observed under an optical microscope (Figure 1), showing polyhedral austenitic grains with frequent twinning.
- Rectangular specimens with dimensions of 15 mm x 40 mm x 1.5 mm were prepared for corrosion resistance testing.
- Some specimens are used in their original state, while others undergo thermal exposure for sensitization.
- Sensitization of the specimens is confirmed through an oxalic acid attack test following ASTM A262 method.
- The specimens are tested for pitting corrosion using potentiodynamic polarization and a 50-day immersion test in a balanced saline solution of Hank.
- Corrosion damage after the immersion test is evaluated through optical microscope observation.
- The electrochemical etching conditions for sensitized specimens are described.
- Some sensitized specimens undergo chemical treatment to remove high-temperature-formed oxides.
- The chemical treatment conditions are indicated.
- The chemical composition of the corrosion solution used is mentioned, as well as the characteristics of the measurement system used for potentiodynamic polarization.
- Potentiodynamic polarization curves are recorded with a scanning rate of 1 mV/s and a specific potential range.
- A 50-day immersion test is also conducted on AISI 316L steel specimens, with weighing before and after the test to assess corrosion damage.
Results and Discussions:
The results demonstrate a correlation between sensitization of AISI 316L steel, decreased corrosion resistance, and the formation of localized corrosion damage. These findings underscore the importance of considering sensitization when using this steel as a biomedical material and highlight the benefits of chemical treatment in improving passive behavior and reducing susceptibility to corrosion.
- Sensitization of AISI 316L steel was confirmed through observation of the microstructure after oxalic acid etching, revealing the formation of grooves along the austenitic grain boundaries.
- Potentiodynamic polarization curves show that sensitization, in combination with high-temperature surface oxides, leads to an active anodic behavior rather than passive behavior.
- Potentiodynamic polarization parameters, such as corrosion potential (Ecorr) and corrosion current density (icorr), were measured and compared for different types of specimens.
- Potentiodynamic polarization curves demonstrate that specimens sensitized with high-temperature surface oxides have reduced corrosion resistance compared to specimens in their original state.
- Chemical treatment with nitric acid and hydrofluoric acid improved the passive behavior of sensitized specimens by removing high-temperature surface oxides.
- Potentiodynamic polarization curves also show a slight decrease in the stability of passive behavior for sensitized specimens compared to specimens in their original state.
- The 50-day immersion tests revealed greater localized corrosion damage on sensitized specimens than on specimens in their original state, particularly along grain boundaries weakened by chromium depletion.
- The calculated average corrosion rates based on mass loss are very low for both types of specimens, but the slight increase in corrosion rate for sensitized specimens is significant in terms of long-term applications, such as surgical implants.
Overall, this article makes a significant contribution to the understanding of pitting corrosion of sensitized AISI 316L stainless steel in biomedical conditions. The presented results and discussions are robust and provide valuable insights for professionals in the field. Considering sensitization of AISI 316L steel is essential in the design and use of long-term surgical implants to ensure their durability and corrosion resistance. I strongly recommend acceptance of this article.

Minor editing of English language required
Author Response
Thank you for your review and the recommendation of our article
Reviewer 4 Report
It is not surprising that sensitized AISI 316L is more susceptible to pitting than that without sensitization. The authors mentioned that the test temperature has importance in consideration for using the AISI 316L as biomaterial for surgical implants, and I agree with the authors. Then, the discussion on the effects of temperature on the pitting behavior of the sensitized samples must be carried out in the research. If not, reader can not understand the importance of test temperature nor whether if test temperature affects experimental results; this manuscript has no new scientific insite.
As the manuscript is very well written, it is a shame that no experiment showing the effects of test temperature is in the manuscript.
Round 2
Reviewer 1 Report
1A) Thank you for the comment. The explanation was added to the Introduction, lines 52 - 59
and to Materials and methods, lines 108-109. In our next research we plan to focus on short
times of heat exposure (simulating of welding conditions).
The question remains then, how reallistic is the HT at 650 C for 40h in order to simulate the welding conditions? Authors did not provide a proper explanation for the HT conditions. It is thus concluded that the used HT parameters are not realistic.
3A) Yes, we suppose that high-temperature oxides are usually removed and surfaces are mostly
polished (electropolishing, plasma polishing…).
It is important to know the process that is followed by the relevant industry in order to follow the same conditions.
4A) Thank you for your comment. We agree that the determination of repassivation potentials
would be useful and would broaden the scope of the discussion. In addition to the fact that one
measurement takes twice as long and the device is very busy, our potentiostat has a problem with
the reproducibility of cyclic curves (large dispersion of repassivation potentials during repeated
measurements). Therefore, we decided to use simple potentiodynamic polarization and, in
addition to it, another independent method (exposure test).
Simple potentiodynamic polarization tests provide very limited information for the resistance of the alloy to localized forms of corrosion. Cyclic polarization would show the presence of negative or positive loops, repassivation potentials etc. As a result the discussion would expand and the results would be more reliable.
If the potentiostat has issues with reproducibility of cyclic polarization curves then questions are raised for the reliability of the forward polarization curves used in this study. Additionally, the length of the experiment or the busy schedule of the lab is not really helping to prove a point.
In conclusion, authors didn`t manage to address most of the comments and suggestions and unfortunately the paper cannot be accepted.
Reviewer 2 Report
COMMENTS:
I checked all the authors' responses to the reviewer's comments and all the issues raised were adequately replied. Therefore, I recommend accepting the article.
Author Response
Thank you for your review
Reviewer 4 Report
Review report on the revision of the materials-2513672.
The authors responded to reviewer's indications, and the manuscript is revised.
recommendation: accept
Author Response
Thank you for your review